# SplashNet: Split-and-Share Encoders for Accurate and Efficient Typing with Surface Electromyography

**Nima Hadidi**[1,2]    **Jason Chan**[3]    **Ebrahim Feghhi**[1,2]    **Jonathan C. Kao**[1,2,3]
University of California, Los Angeles
nhadidi@g.ucla.edu

## Abstract

Surface electromyography (sEMG) at the wrists could enable natural, keyboard-free text entry, yet the state-of-the-art `emg2qwerty` baseline still misrecognizes 51.8% of characters zero-shot on unseen users and 7.0% after user-specific fine-tuning. We trace much of these errors to mismatched cross-user signal statistics, fragile reliance on high-order feature dependencies, and the absence of architectural inductive biases aligned with the bilateral nature of typing. To address these issues, we introduce three simple modifications: (i) Rolling Time Normalization which adaptively aligns input distributions across users; (ii) Aggressive Channel Masking, which encourages reliance on low-order feature combinations more likely to generalize across users; and (iii) a Split-and-Share encoder that processes each hand independently with weight-shared streams to reflect the bilateral symmetry of the neuromuscular system. Combined with a five-fold reduction in spectral resolution (33→6 frequency bands), these components yield a compact Split-and-Share model, SplashNet-mini, which uses only ¼ the parameters and 0.6× the FLOPs of the baseline while reducing character error rate (CER) to 36.4% zero-shot and 5.9% after fine-tuning. An upscaled variant, SplashNet (½ parameters, 1.15× FLOPs of the baseline), further lowers error to 35.7% and 5.5%, representing 31% and 21% relative improvements in the zero-shot and finetuned settings, respectively. SplashNet therefore establishes a new state-of-the-art without requiring additional data.

## 1 Introduction

Translating neuromuscular signals into typed text is a novel but rapidly developing area at the intersection of human-computer interaction and machine learning. The `emg2qwerty` dataset (14) is the first large-scale benchmark specifically for wrist EMG-based touch typing. It contains over 346 hours of data from 108 users typing sentences while wearing electrode bands on each wrist. This dataset was motivated by the promise of wrist EMG as an always-available input modality for AR/VR glasses and other scenarios where traditional keyboards are impractical. In parallel, related benchmarks like `emg2pose` have been developed for EMG-based hand pose estimation (13), reflecting a broad interest in neuromotor interfaces for controlling virtual objects, robots, or text entry. Early explorations of EMG for text input date back at least two decades: for example, Yu et al. (22) demonstrated an "EMG keyboard" that allowed a forearm amputee to input characters via muscle signals. These efforts were limited in scale and accuracy, but they proved the principle that muscle activity alone can convey typing information. Today, with much larger datasets and deep learning models, the accuracy of EMG-based typing has greatly improved (2; 14; 13), making it a viable research direction for assistive technology and next-generation user interfaces.

---

[1] Neuroscience Interdepartmental Program, [2] Department of Electrical and Computer Engineering, [3] Department of Computer Science. Code and checkpoints available at github.com/nhadidi/SplashNet

39th Conference on Neural Information Processing Systems (NeurIPS 2025).

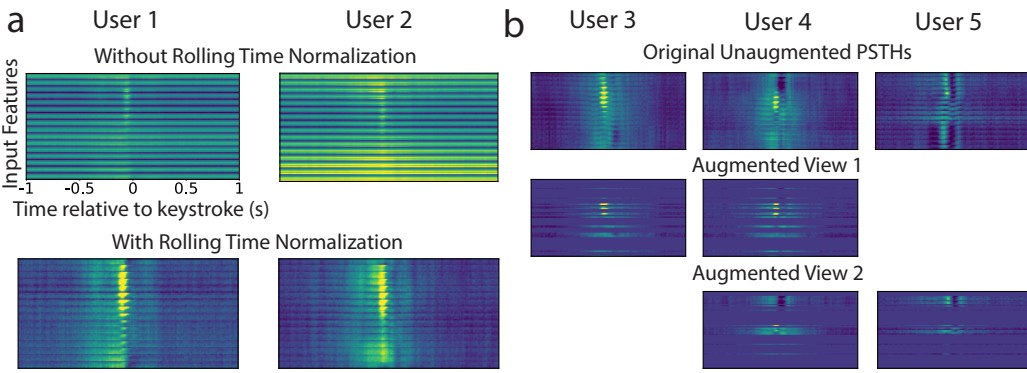

Figure 1: **a)** Top row: Peri-stimulus time histograms (PSTHs) for the **"e"** key with (top) and without (bottom) RTN for two users. Each PSTH shows the spectral features derived from the left hand, with spectrograms from the 16 electrodes concatenated together. RTN mitigates the significant differences in across-user feature scale and bias. **b)** Top Row: PSTHs for the **"e"** key from 3 training users. Note that some of User 4's features show similar patterns to User 3, while others show similar patterns to User 5. ACM isolates small feature combinations, which are more often shared across users.

Although promising, EMG interfaces face critical challenges in generalization. EMG interfaces suffer from substantial **domain shift** across users and sessions: anatomy, electrode placement, fatigue, and day-to-day physiology all alter the signal for the same action (14). The emg2qwerty benchmark probes this shift with (i) a *zero-shot* test, where one trains on many users and tests on a new one, and (ii) a *personalization* test that finetunes on a few target-user samples. Their baseline ASR-style system (CNN on spectrograms + LM decoding) misrecognized $51.8\%$ of characters zero-shot, but finetuning cut errors to approximately $7\%$, showing that most residual errors are systematic, user-specific variations. Yet even large training pools struggle: Sivakumar et al. (14) estimate $\mathcal{O}(10^3)$ users are needed for robust out-of-the-box performance, a trend echoed by CTRL-Labs' gains from massive data (2).

While scaling may significantly increase the performance of EMG interfaces, collecting these large-scale datasets is time-consuming and human-resource expensive. We instead present a simpler and complementary path towards improving sEMG generalization. Our key contribution is to first identify limitations in sEMG data that form obstacles to zero-shot generalization. We then propose *simple*, *causal*, and *computationally inexpensive* modifications to address these limitations, resulting in models that are more invariant to user/session differences. Our rationale for pursuing causal and computationally inexpensive modifications is to enable sEMG decoders to work on-device, an important consideration for future wrist EMG devices.

## 2 Insights into improving sEMG decoding

We summarily make three insights, and subsequently, three simple modifications to sEMG decoding that substantially improve zero-shot and finetuning performance while reducing computational costs.

**Insight 1: sEMG features should be *causally* normalized *per session*.** sEMG amplitudes routinely differ by an order of magnitude across participants or sessions (3; 8). Figure 1a highlights this disparity: the same channels exhibit both higher baselines and greater variance for User 2 than for User 1. Hence, EMG signals should obviously be normalized between users. However, the method of normalization is critical. Sivakumar et al. (14) apply batch normalization that computes mean and variance over the mini-batch, time, and per-electrode frequency bins. Because each mini-batch mixes data from many users and sessions, these statistics fail to place features from different users/sessions in a common space during training and remain fixed at inference. We instead employ **Rolling Time Normalization (RTN)**, a causal z-scoring of every input feature that updates its mean and variance online. Though simple, this mitigates the scale and shift differences *across users*, visualized in Figure 1a. Further, RTN requires no calibration and adds virtually no latency for causal, real-time inference. We empirically find this significantly improves zero-shot and finetuned performance.

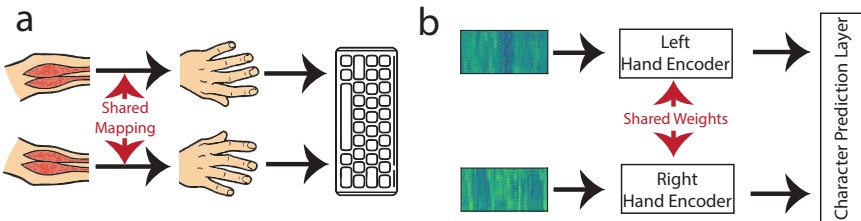

Figure 2: **a)** The bilateral structure of keyboard typing. **b)** The *Split-and-Share* macro-architecture.

**Insight 2: sEMG feature *subsets* may be conserved across users.** Through visualizing sEMG activity, we empirically find that while two typists rarely share identical high-dimensional EMG signatures for the same keystroke, they often share *subsets* of electrodes or frequency bands that behave similarly. A representative example is shown in Figure 1b, where we present the PSTHs of sEMG activity from three users. While the full set of features looks fairly different across users, randomly selected subsets of features have a greater chance of being similar between some users (Augmented View 1 and 2 in Figure 1b). Decoding strategies that hinge on these low-order combinations therefore stand a better chance of transferring across users. While there are many ways to encourage this, we focus on substantially modifying the hyperparameters of dataset augmentation to achieve this, incurring no test-time computational expense. In particular, we do **Aggressive Channel Masking (ACM)** that on average zeros out more than half of the electrodes and removes broad spectral chunks from the rest. ACM encourages the model to rely on smaller, more universal feature sets (Figure 1b), which empirically improves zero-shot generalization. We emphasize that this 'low-order similarity' perspective is a working hypothesis rather than a definitive statement about the structure of EMG representations: the empirical improvements we observe are consistent with this interpretation but do not prove it.

**Insight 3: an inductive bias for bilateral typing.** Keyboard typing is fundamentally bilateral: each keystroke is driven solely by the muscles of its own wrist, while the neuromuscular mapping from activation to finger motion is almost mirror-symmetric across hands. We therefore propose a **Split-and-Share** architecture that accounts for these two facts in the model. Separate, but *weight-shared*, subnets encode left- and right-hand sEMG streams in parallel, converging only at the final linear layer where a single vocabulary-level softmax decodes the character (Figure 2). This enforces hand-specific locality, but prevents the encoder from entangling spurious cross-hand correlations. Further, re-use of parameters reduces resource demands: the compact Split-and-Share model uses $0.25\times$ the parameters and approximately $60\%$ of the FLOPs of a conventional joint-hand encoder while matching zero-shot accuracy.

Summarily, our new, lightweight architecture built on simple modifications from these three insights reduces zero-shot generalization by $31\%$ and finetuning performance by $21\%$ over the prior state-of-the-art in Sivakumar et al. (14). Together, we both increase performance while reducing computational cost, providing a more feasible path towards on-device, performant computation for sEMG interfaces.

## 3   Related Works

**Per-Session Normalization in EMG**: The EMG and brain-computer interface (BCI) literature commonly normalize input signals prior to decoding to account for variability. For instance, prior EMG studies expressed amplitudes as a percentage of a reference contraction (maximum voluntary contraction) or applied z-score normalization based on a calibration recording (15). These normalizations enable comparisons across muscles and subjects (3; 8), and several works have normalized EMG features per session for machine learning pipelines (15; 10; 7; 20; 16; 6). In BCI literature, Willett et al. (19) show that continuously updating z-score statistics on spiking features within each session is essential for their high-performance speech neuroprosthesis, and Jarosiewicz et al. (5) used online bias correction to improve a point-and-click BCI. As described in Section 2, an important insight is not whether sEMG activity should be normalized, but *how it is normalized*. We ultimately find RTN significantly outperforms batch normalization.

**Masking-Based Regularization for sEMG**: SpecAugment, first introduced for speech recognition, has recently been ported to EMG decoding pipelines (12; 14). The version used by Sivakumar et al. (14) is relatively mild: on average only $\sim 6\%$ of inputs are masked per training sample, with a given electrode's spectrogram never being masked across more than $\sim 24\%$ of the frequencies. ACM corresponds to hyperparameters that are intentionally harsher, with at least half of the electrodes completely blanked in most mini-batches and $\sim 55\%$ of input features masked on average. This encourages the model to infer keystrokes from small, overlapping subsets of channels, discouraging brittle reliance on the full spatiotemporal feature pattern.

**Multi-stream architectures and weight sharing**: The concept of processing multiple information streams independently before later integration has been established in Automatic Speech Recognition (ASR), where multi-stream systems effectively handle speech at various resolutions or from multiple arrays (9). Similarly, in Sign Language Recognition (SLR), previous works have implemented separate processing streams for each hand (11; 17). Leveraging anatomical symmetry through weight sharing has shown success in pose estimation (21) and SLR (17). In the domain of EMG-based hand-pose recognition, Salter et al. (13) combined data from both hands into a unified dataset for single-model training, though this addresses an inherently unimanual task. For bimanual keystroke decoding, past methods have processed signals from both hands jointly (14; 18; 1). Our Split-and-Share architecture keeps the two wrists separate to respect biomechanics, yet ties the weights, effectively training a single-hand encoder with data from both hands. To our knowledge, this has not been explored in EMG keyboard decoding.

## 4 Methods

### 4.1 Reduced Spectral Granularity

Let $x_c(t)$ denote the raw 2 kHz EMG signal from electrode $c$. Following Sivakumar et al. (14) we first form a log-power spectrogram

$$S_c(t, f) = \log_{10}\Big(\big|\mathrm{STFT}(x_c)\big|^2_{t,f} + 10^{-6}\Big), \qquad f = 1, \ldots, 33,$$

using a 64-point FFT (33 linearly spaced bins up to 1 kHz) and a hop size of 16. We instead aggregate the 33 bins into six broader, roughly log-spaced bands, using the same frequency ranges as in (2). We refer to this aggregation as reduced spectral granularity (RSG).

$$\mathcal{B}_1 = [31.25, 62.5], \ \mathcal{B}_2 = [62.5, 125], \ \mathcal{B}_3 = [125, 250],$$

$$\mathcal{B}_4 = [250, 375], \ \mathcal{B}_5 = [375, 687.5], \ \mathcal{B}_6 = [687.5, 1000] \text{ Hz.}$$

For each electrode the reduced spectrogram is obtained by

$$R_c(t, b) = \sum_{f \in \mathcal{B}_b} S_c(t, f), \qquad b = 1, \ldots, 6.$$

Hence, we reduce the spectral dimensionality of each electrode from 33 to 6. This represents an over five-fold reduction in features, and empirically equals or improves performance. This also leads SpecAugment to mask significantly more frequencies, even prior to ACM.

### 4.2 Rolling Time Normalization

Sivakumar et al. (14) normalizes each electrode's spectrogram with batch-norm over the entire mini-batch, frequency, and time axes. This collapses all features recorded on an electrode into a single distribution and, crucially, re-uses statistics gathered during training at inference time—an issue when the test session (or user) drifts from the training distribution. RTN replaces this with a *causal, per-feature* normalizer that is computed independently for every sample, band, electrode channel and frequency bin. Let

$$x_{t,n,b,c,f} \in \mathbb{R}, \qquad t = 0 \ldots T-1, \ n = 0 \ldots N-1,$$

denote the log-power value at time-step $t$, mini-batch index $n$, band $b$, electrode $c$ and frequency bin $f$. RTN maintains cumulative statistics

$$\mu_t = \frac{1}{t+1} \sum_{s=0}^{t} x_{s,n,b,c,f}, \qquad \sigma_t = \sqrt{\left(\frac{1}{t+1}\sum_{s=0}^{t} x^2_{s,n,b,c,f}\right) - \mu_t^2 + \varepsilon},$$

which are cheap to update via running sums in streaming settings. During a warm-up period of $T_w = 125$ frames (the first 1 second) the statistics are frozen to those computed over the entire warm-up window. The normalized output is

$$\hat{x}_{t,n,b,c,f} = \frac{x_{t,n,b,c,f} - \tilde{\mu}_t}{\tilde{\sigma}_t},$$

where $\tilde{\mu}_t$ and $\tilde{\sigma}_t$ denote the warm-up-frozen statistics for $t < T_w$ and the cumulative statistics above for $t \geq T_w$. RTN adapts continuously to session-specific non-stationarities, avoids batch-level or training-level statistics, and is causal.

### 4.3 Aggressive Channel Masking

We apply ACM on the reduced spectral granularity features with $B = 6$ bands. We mask inputs in $2/3$ of mini-batches. For each mini-batch we draw the number of frequency masks, $n_f \sim \text{Unif}\{0, 1, 2\}$. Each mask is then sampled independently for every electrode and sample in the batch:

1. Width $w \sim \text{Unif}\{0, \ldots, f_{\max} - 1\}$, with $f_{\max}=12$. The implementation clamps the width to the number of bands, $w \leftarrow \min(w, B)$. Thus the probability that a single mask erases the entire electrode is $\Pr[w = B] = \frac{f_{\max} - B}{f_{\max}} = 0.5$.

2. Start index $f_0 \sim \text{Unif}\{0, \ldots, B - w\}$.

3. Set $X_{t,c,f_0:f_0+w-1} \leftarrow 0$ for all timesteps $t$.

An individual mask already removes a channel with probability 0.5, while two masks remove a channel with probability 0.75. Remaining electrodes lose large spectrally-coherent chunks. By forcing the network to succeed even when so many electrodes are fully masked, the augmentation discourages brittle high-order feature dependencies and promotes low-order motifs that transfer across users.

### 4.4 EMG Encoder Architectures

Following spectrogram normalization, all architectures apply a two-stage encoder composed of a *Rotation-Invariant MLP* and a *Time-Depth Separable Convolution* (TDSConv) stack, with a linear character prediction layer following the last TDSConv block. These architectures were first presented in Sivakumar et al. (14) and are reproduced here for completeness.

**Rotation Invariant MLP.** Let $\mathbf{x} \in \mathbb{R}^{T \times N \times B \times C \times F}$ denote the input, where $T$ is the number of time steps, $N$ is the batch size, $B = 2$ denotes the number of bands (hands), $C = 16$ is the number of electrode channels per band, and $F$ is the number of spectral frequency bins. Each band is passed through a Rotation-Invariant MLP, which applies a multi-layer perceptron to rotated versions of the electrode channels to ensure robustness to cyclic spatial shifts:

$$\mathbf{h}_b = \frac{1}{|\mathcal{O}|} \sum_{o \in \mathcal{O}} \text{MLP}(\text{roll}(\mathbf{x}_b, o)), \quad \mathcal{O} = \{-1, 0, 1\},$$

where each rotated input is flattened across $C \times F$ before the MLP is applied. This results in an embedding $\mathbf{h}_b \in \mathbb{R}^{T \times N \times D}$ per band. Concatenating the left and right hand features yields a total input of shape $\mathbf{h} \in \mathbb{R}^{T \times N \times 2D}$ to the TDSConv blocks in the *baseline model*, whereas each hand is processed independently in the *Split-only* and *Split-and-Share* variants.

**TDSConv block architecture.** The TDSConv stack (4) is composed of alternating temporal convolution and fully connected blocks. Each convolutional block first reshapes the input from $\mathbb{R}^{T \times N \times D}$ to $\mathbb{R}^{N \times K \times H \times T}$, where $D = K \cdot H$ denotes the total feature dimension, $K$ is the number of convolution channels, and $H$ is the per-channel hidden width. A 2-D convolution with kernel size $1 \times w$ is then applied along the time axis:

$$\mathbf{z}[n, k, h, t] = \sum_{i=0}^{w-1} \sum_{k'=1}^{K} \theta_{k,k',i} \, \mathbf{h}[n, k', h, t - i], \qquad \theta \in \mathbb{R}^{K \times K \times 1 \times w}.$$

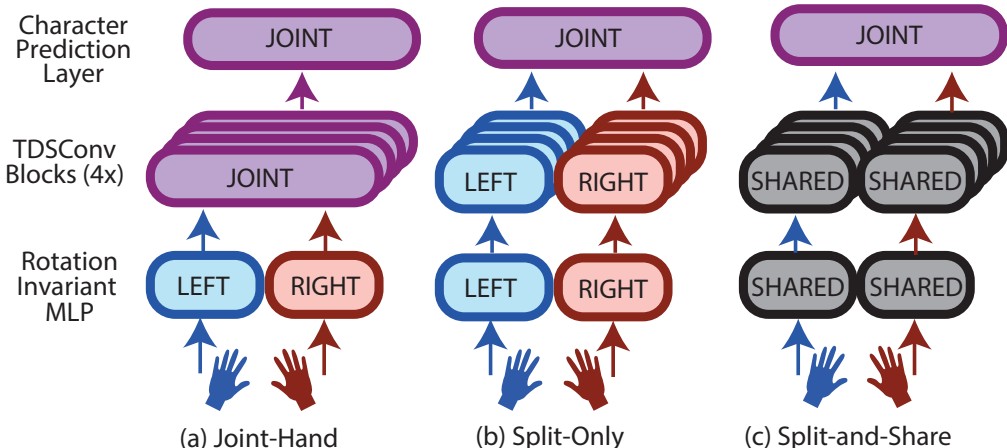

Figure 3: EMG encoder architectures. Left (blue) and right (red) hand specific modules only process inputs from a single hand, with hand-specific weights. Joint (purple) modules jointly process inputs from both hands. Shared (gray) modules process inputs from either hand using identical weights. **a)** *Joint-Hand* baseline architecture of Sivakumar et al. (14), **b)** *Split-only* architecture, in which hand-specific modules process signals from each hand separately. **c)** *Split-and-Share* architecture, where shared-weight modules process signals from each hand separately.

Because the kernel height is 1, the same weights are reused at every hidden-width position $h$. The convolution still mixes all $K$ input channels to produce each of the $K$ output channels, giving the layer $K^2 w$ parameters. The output is passed through ReLU, summed with the residual, and normalized using LayerNorm. The subsequent fully connected block consists of two linear layers with a ReLU in between and a residual connection:

$$\mathbf{z}_{\text{fc}} = \text{LayerNorm}(\text{FC}_2(\text{ReLU}(\text{FC}_1(\mathbf{z}_{\text{conv}}))) + \mathbf{z}_{\text{conv}}).$$

**Architecure variants.** In the architecture of Sivakumar et al. (14), which we refer to as the *Joint-Hand* architecture, both hands are processed jointly following the Rotation-Invariant MLP, yielding a post-MLP embedding of $2D = 768$. The TDSConv stack operates with this full dimensionality throughout. In contrast, both the *Split-only* and *Split-and-Share* models operate on embeddings of $D = 384$ per hand, and apply the TDSConv blocks separately to each hand's input. This reduces the width of all fully connected layers from $768 \times 768$ to $384 \times 384$, leading to a four-fold reduction in parameters per FC layer, while preserving expressiveness by keeping the number of convolution channels $K = 24$ and kernel width $w = 32$ unchanged. Despite duplicating the encoder, the *Split-only* model has lower total FLOPs—approximately 60% of the baseline—due to the narrower per-hand embeddings. The *Split-and-Share* model uses the same dual-stream structure but shares all encoder weights between hands. This model has roughly half the parameters of the *Split-only* model, since almost all parameters (excluding the final prediction layer) are shared between the two encoders. Notably, the *Split-only* and *Split-and-Share* models have the same FLOPs.

To evaluate whether representational capacity might further enhance the strong performance of the *Split-and-Share* architecture, we also test an *Upscaled Split-and-Share* variant. This model increases the per-hand embedding size to $D = 528$ and expands the number of convolution channels in the final two TDS blocks from 24 to 48. Nonetheless, the total number of parameters remains about half that of the baseline, and the FLOPs are only modestly higher (about 15%). We refer to the *Upscaled Split-and-Share* model as *SplashNet* and the smaller *Split-and-Share* model as *SplashNet-mini*

### 4.5 Backspace-aware beam search with 6-gram Character LM

For decoding we adopt the backspace-aware beam search of Sivakumar et al. (14), which keeps the 50 most-probable CTC label prefixes at each frame. Blank transitions simply update the score of an unchanged prefix, while the highest-scoring non-blank characters extend each hypothesis and are re-ranked with a 6-gram character LM prior. When a backspace symbol appears, the algorithm retracts the last language-model contribution, letting deletions correct earlier mistakes without extra

Table 1: Zero-shot CER (%, mean $\pm$ s.d. across participants), GFLOPs, and parameters. Columns in gray correspond to training domain validation results, which are reported for transparency but not used as indicators of generalization.

| Method | Train domain val | Other domain val | Test domain val | Test domain test | GFLOPs (30 s) | Params |
|---|---|---|---|---|---|---|
| Sivakumar et al. 2024 | 12.14 | 72.07 | $52.10 \pm 5.54$ | $51.78 \pm 4.61$ | 61.61 | 5.29M |
| + RSG | 13.52 | 67.48 | $47.26 \pm 5.26$ | $47.18 \pm 5.19$ | 54.15 | 4.96M |
| + RTN | 13.09 | 61.95 | $39.49 \pm 7.45$ | $39.15 \pm 6.20$ | 54.15 | 4.96M |
| + ACM | 23.47 | 63.08 | $42.62 \pm 7.18$ | $42.62 \pm 7.10$ | 54.15 | 4.96M |
| + RTN + ACM | 21.71 | 58.85 | $36.41 \pm 7.21$ | $36.42 \pm 7.11$ | 54.15 | 4.96M |
| + Split | 23.93 | 58.64 | $37.28 \pm 6.91$ | $37.37 \pm 7.34$ | 36.84 | 2.68M |
| + Share (*SplashNet-mini*) | 26.44 | 58.20 | $36.46 \pm 7.09$ | $36.41 \pm 7.30$ | 36.84 | 1.38M |
| + Upscale (*SplashNet*) | 20.59 | **56.95** | $\mathbf{35.49 \pm 7.56}$ | $\mathbf{35.67 \pm 6.79}$ | 71.38 | 2.58M |

passes. After the final frame any open LM context is closed, and the best-scoring prefix provides the decoded keystroke sequence. All beam-search parameters are the same as in Sivakumar et al. (14)

## 4.6 Dataset Splits

The official *emg2qwerty* protocol introduced by Sivakumar et al. (14) assigns 100 participants to the training pool and eight to a held-out test pool. For the 96 training participants with $\geq 4$ recording sessions, every session except the final two is used to train *generic* models, while those last two sessions form the authors' validation set; models are selected on this set before being evaluated on the eight held-out participants.

Because the validation data come from the same individuals seen during training, this procedure encourages models that memorize participant-specific idiosyncrasies rather than ones that generalize across users. To illustrate this effect while limiting computational cost, we evaluate training domain validation performance using the validation sessions from 18 of the 96 training users, which we term the *training domain validation set*. The same 18 users are also used in Appendix A.8, where they are fully held out from training to provide a more appropriate validation signal.

To obtain a validation signal that better predicts zero-shot performance in the main experiments, we instead exploit the four training-pool participants who recorded $< 4$ sessions and were therefore excluded from model fitting in the original setup. We take the final session of each of these four participants and combine them into what we term the *other domain validation set*. These participants exhibit noisier EMG data than those in the official test pool, providing a harder proxy for real-world generalization. Nonetheless, they provide a way of validating zero-shot generalization without requiring a complete restructuring of the official dataset splits.

## 5 Results

### 5.1 Zero-shot model performance

Before turning to the main cross-user generalization results, we first highlight a key discrepancy between training domain validation and held-out user evaluation. As shown in Table 1, the CERs on the training domain validation set are not only lower, but the relative ranking of models follows a markedly different pattern than on held-out user evaluation sets. This illustrates why training domain validation provides a misleading signal for cross-user generalization and should not be used for model selection. In Appendix A.8, we further show that when these same 18 users are held out entirely, their CERs rise substantially and align closely with the held-out evaluation trends shown here.

Our analysis centers on beam-search decoding using the same 6-gram character LM as in Sivakumar et al. (14); corresponding greedy-decoding results appear in Appendix A.3. All reported p-values were obtained from one-tailed paired t-tests across participants.

Table 2: Finetuned CER (%, mean ± s.d. across participants), GFLOPs, and parameters.

| Method | Test domain val | Test domain test | GFLOPs (30 s) | Params |
|---|---|---|---|---|
| Sivakumar et al. 2024 | 8.31 ± 3.19 | 6.95 ± 3.61 | 61.61 | 5.29M |
| + RSG | 6.70 ± 3.22 | 6.92 ± 3.79 | 54.15 | 4.96M |
| + RTN | 6.47 ± 2.92 | 6.63 ± 3.58 | 54.15 | 4.96M |
| + ACM | 7.18 ± 3.33 | 7.45 ± 3.87 | 54.15 | 4.96M |
| + RTN + ACM | 6.55 ± 2.91 | 6.53 ± 3.27 | 54.15 | 4.96M |
| + Split | 6.04 ± 2.81 | 6.10 ± 3.32 | 36.84 | 2.68M |
| + Share (*SplashNet-mini*) | 6.13 ± 2.96 | 5.87 ± 3.04 | 36.84 | 1.38M |
| + FT Unshare | 5.85 ± 2.83 | 5.96 ± 3.28 | 36.84 | 2.68M |
| + Upscale (*SplashNet*) | **5.46 ± 2.60** | **5.51 ± 2.81** | 71.38 | 2.58M |
| + FT Unshare | 5.57 ± 2.65 | 5.67 ± 2.97 | 71.38 | 5.06M |

First, we apply RSG to replace the 33-bin spectrogram with six coarser frequency bands. RSG leads to a modest but consistent improvement in CER from $51.78\%$ to $47.18\%$ ($p = 7e\text{-}5$). Part of this performance improvement may stem from the implicit amplification of frequency masking under SpecAugment (due to the lower spectral resolution).

Second, we replace batch-level normalization with RTN. RTN significantly improves zero-shot generalization ($p = 6e\text{-}4$), reducing the CER to $39.15\%$. This suggests that input scale and shift differences across users are a primary obstacle to cross-user generalization in EMG decoding.

Third, we apply ACM, which encourages the model to rely on lower-order combinations of input features. Without RTN, ACM reduces zero-shot CER to $42.6\%$ ($p = .02$ vs. +RSG only). Combined with RTN, ACM reduces zero-shot CER to $36.42\%$ ($p = 5e\text{-}3$ vs. +RSG+RTN), providing a strong *Joint-Hand* baseline.

Fourth, we explore architectural modifications that better reflect the causal and bilateral structure of EMG typing. We evaluate a *Split-only* model that encodes each hand separately without parameter sharing, achieving a CER of $37.37\%$. Despite having only half the parameters and $66\%$ of the FLOPs of our *Joint-Hand* baseline, this model performs competitively, though its lack of shared parameters may limit data efficiency. We then evaluate *SplashNet-mini*, where both hands are encoded via identical weight-shared encoders. This model achieves a $36.41\%$ CER—on par with the *Joint-Hand* baseline—while using just a quarter of its parameters and $66\%$ of the FLOPs. Finally, by increasing the embedding width and expanding the final convolutional layers, we create *SplashNet*, with similar FLOPs to the baseline of Sivakumar et al. (14) but still half the parameters. This yields a further improvement, reducing the CER to $35.67\%$ ($p = .049$ vs. SplashNet-mini).

Together, these results demonstrate that a combination of architectural priors, per-session normalization and principled regularization can significantly improve zero-shot EMG decoding.

## 5.2 Finetuned Model Performance

We next evaluate the performance of models finetuned on user-specific data. As in Sivakumar et al. (14), we maintain identical training hyperparameters during both generic pretraining and finetuning. While this simplifies the analysis, we note that the optimal hyperparameters for finetuning likely differ from those used during pretraining, and further gains may be achievable through phase-specific hyperparameter tuning. We also note that, because each recording session involved doffing and re-donning the wristbands, the finetuning experiments inherently probe generalization across electrode placements in different sessions.

We report all results using beam search with an external character-level language model (LM), which remains the standard for achieving state-of-the-art performance in CTC-based decoding pipelines. While our models consistently outperform the baseline of Sivakumar et al. (14) under beam search, we observe slightly worse performance under greedy decoding. We attribute this in part to the more aggressive channel masking induced by our reduced spectral representation (§4), which we discuss further in Appendix A.3.

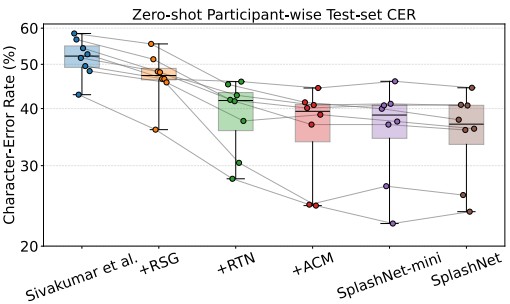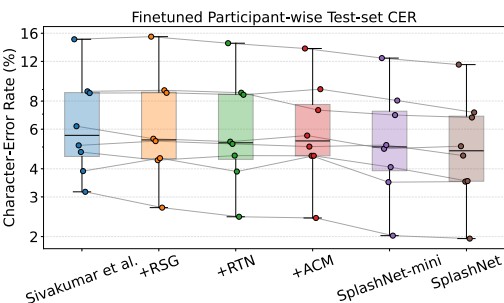

Figure 4: Zero-shot and finetuned CER distribution across users. Each of the 8 test users are represented by a dot, with lines connecting the same user across models. Boxplots depict median and interquartile ranges. Our methods improve performance for all participants relative to the baseline of Sivakumar et al. (14), with some participants showing very large improvements: two users reach CER between 20-30% in the zero-shot setting, and one user attains a CER below 2% when finetuned.

First, we assess whether the same methods that improved zero-shot generalization also enhance performance in the finetuned setting. Using RSG alone yields similar performance to the baseline of Sivakumar et al. (14), with a CER of 6.91%, while adding RTN yields a significantly improved CER of 6.63% ($p = .028$ vs. +RSG). Interestingly, applying ACM in isolation—without RTN—leads to worse performance than the baseline of Sivakumar et al. (14), whereas the *Joint-Hand* baseline model, with both ACM and RTN, matches the model with RTN alone (6.53% CER). This suggests that masking-induced variability may hinder learning when the model lacks an appropriate normalization strategy to stabilize the input feature space. Moreover, unlike the generic case, finetuned models may rely more on higher-order feature correlations that are relatively stable across sessions for the same user, diminishing the benefits of aggressive masking.

We next examine architectural changes that explicitly encode inductive biases about the bilateral and causal structure of EMG typing. Both the *Split-only* and *SplashNet-mini* models yield substantial improvements over the *Joint-Hand* baseline, with CERs of 6.10% and 5.87% ($p = 6e\text{-}3$ and $p = 2e\text{-}3$, respectively). For the latter, we evaluate two strategies during finetuning: either maintaining shared weights or duplicating them to allow separate adaptation per hand. Interestingly, both strategies yield similar performance, suggesting that hand-specific encoder weights are unnecessary even in the finetuned setting, although it is possible that unsharing weights could become advantageous with more user-specfic data for finetuning.

Finally, we evaluate the *SplashNet* model. Again, we do not find any benefit from unsharing the weights during finetuning. With the weights kept shared, *SplashNet* achieves the best performance overall with a CER of 5.51% ($p = .02$ vs. *SplashNet-mini*), establishing a new state-of-the-art for user-specific EMG keystroke decoding in this benchmark.

## 6 Discussion

The central contribution of this work is instilling simple, well-motivated priors through preprocessing, augmentation and architecture that can close a surprising fraction of the generalization gap in wrist-EMG typing. SplashNet-mini and SplashNet both achieve large absolute and relative CER reductions (-15.4 pp / -31.1% zero-shot; -1.44 pp / -20.7% after fine-tuning) while cutting parameters to ¼–½ of the baseline and FLOPs to 0.6–1.15×. These gains are on par with (and complementary to) the $\sim 25\%$ error reduction CTRL-Labs at Reality Labs et al. (2) reported from *doubling* dataset size in a handwriting task, suggesting that principled inductive biases are as potent as raw data scaling for sEMG.

A practical ambition articulated in Sivakumar et al. (14) is to run the entire decoder on the wristbands themselves, thereby mitigating concerns around latency, privacy, and robustness to Bluetooth interference. Achieving on-band inference will require an architecture that is not only light-weight but also fully split—able to process and output keystrokes from each wrist independently, without any cross-hand coupling. SplashNet moves part-way toward this goal by duplicating (and sharing) the encoder streams, yet it still merges information at the final linear output layer, so each wrist must

communicate its embeddings to the other. Removing this last dependency, or replacing it with a low-bandwidth handshake, remains an open engineering challenge and a fruitful direction for future model architecture work. Another promising direction is exploring hybrid cross-hand architectures that retain split-and-share components but introduce limited, structured interactions between the hands, potentially offering a middle ground between the Split-and-Share and Joint-Hand extremes.

Beyond these concrete results, our study points to several avenues. *Normalization:* RTN is a single-pass causal z-score; richer adaptive schemes (momentum updates, learnable affine transforms, or brief self-supervised calibration) may yield further robustness. *Structured masking:* ACM already improves zero-shot CER, but spatially contiguous electrode "drop-blocks", global time masks, or anatomical adjacency priors could guide the network toward even more transferable features. *Model scale and self-supervision:* SplashNet shows that capacity can be reinvested profitably once good priors are in place; coupling larger split-and-share encoders with masked-prediction pre-training may unlock still-higher accuracy without additional labels.

Two limitations deserve emphasis. First, our models are developed for the practical beam-search + LM pipeline; under pure greedy decoding, the *Joint-Hand* baseline scores the best on unseen users, and ACM dents finetuned accuracy (details and discussion in Appendix A.3). This likely reflects the fact that the *Joint-Hand* baseline can exploit cross-hand co-articulatory patterns, which tend to correlate strongly with character bi- and tri-grams. These weak statistical regularities effectively act as an implicit language model under greedy decoding. By contrast, Split-and-Share models do not have access to these bilateral dependencies, and ACM further suppresses within-user cues by enforcing lower-order feature reliance. Once an external 6-gram LM is applied, however, its much stronger prior readily compensates for both the lost co-articulatory signal of the Split-and-Share models and the reduced within-user discriminability induced by ACM, thereby revealing the underlying generalization advantage of our approach. Second, all experiments use healthy participants; the extent to which our priors transfer to populations with motor or limb differences (where electrode placement and muscle recruitment differ markedly) remains to be tested.

In sum, we show that the long-standing vision of "keyboard-quality" EMG typing can be advanced not only by *more data* but also by *better assumptions*. By pairing causal normalization, aggressive yet structured regularization, and a symmetry-aware encoder, we make significant strides towards truly *out-of-the-box* wrist-sEMG typing—paving the way for real-world assistive and AR/VR interfaces.

## 7 Acknowledgments

This work was supported by the following awards to JCK: NSF CAREER 1943467, NIH DP2NS122037, NIH R01NS121097. The authors would also like to thank Andrea Ortone for carefully checking the FLOPs calculations and providing helpful feedback.

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

## A Technical Appendices and Supplementary Material

### A.1 Training Details

All models were trained for 150 epochs with the hyperparameter configuration of Sivakumar et al. (14). we used the same optimizer (Adam), the same LR schedule (linear warmup from 1e-8 to 1e-3 for first 10 epochs, followed with cosine annealing to 1e-6 until epoch 150), and the same augmentations. These included a rotation augmentation, which shifts all electrodes on each band one electrode to the left or right for each training sample, and a temporal alignment jitter augmentation, which jitters the EMG signals from each hand by a maximum offset of 60 ms. We only made two modifications to hyperparameters:

1. **Input window length.** We employ 16 s training windows, rather than the 4 s clips used by Sivakumar et al. (14). This was done to speed up training.

2. **SpecAugment settings.**
   - *With ACM.* We raise the maximum frequency mask width to 12 and disable time masking.
   - *Without ACM.* We retain the original SpecAugment parameters of Sivakumar et al. (14) (maximum frequency mask width = 4; up to 3 time masks per electrode per sample, each masking out up to 200 ms).

### A.2 Evaluation Details

All validation and test sessions were loaded as a single sample (i.e. with batch size of 1). By comparison, Sivakumar et al. (14) loaded validation (but not test) sessions as shortened chunks. Although this is unlikely to have affected their greedy decoding results as their architecture (and ours) has a receptive field of only 1 second, this might have affected the validation performance they reported with an external LM, which would lose the character history between each chunk and therefore make poorer predictions. This might explain why the validation CER they report with beam-search (8.31% CER) is conspicuously higher than their test performance (6.95% CER) compared to the gaps we see between validation and test sessions.

## A.3 Greedy vs. Beam-Search Decoding

Our main text focuses on beam-search decoding with an external 6-gram language model (LM), because this is the configuration most relevant to realistic, latency-constrained deployment. For completeness, we also report *greedy* CTC decoding results, i.e. decoding without any LM.

Table 3: Zero-shot CER (%, mean across participants). Columns in gray correspond to training domain validation results, which are reported for transparency but not used as indicators of generalization. Note that the training domain validation results shown here with greedy decoding were computed over all 96 training users; the beam search decoding results are computed on the 18 user subset.

| Method | Train domain val (greedy) | Train domain val (beam) | Other domain val (greedy) | Other domain val (beam) | Test domain val (greedy) | Test domain val (beam) | Test domain test (greedy) | Test domain test (beam) |
|---|---|---|---|---|---|---|---|---|
| Sivakumar et al. 2024 | 22.51 | 12.14 | 72.44 | 72.07 | 55.57 | 52.10 | 55.38 | 51.78 |
| + RSG | 23.59 | 13.52 | 68.55 | 67.48 | 52.24 | 47.26 | 52.27 | 47.18 |
| + RTN | 23.49 | 13.09 | 64.19 | 61.95 | 46.31 | 39.49 | 46.07 | 39.15 |
| + ACM | 34.77 | 23.47 | 65.60 | 63.08 | 48.87 | 42.62 | 49.23 | 42.62 |
| + RTN + ACM | 32.85 | 21.71 | 61.82 | 58.85 | 43.72 | 36.41 | 43.80 | 36.42 |
| + Split | 35.59 | 23.93 | 61.74 | 58.64 | 45.73 | 37.28 | 45.69 | 37.37 |
| + Share (*SplashNet-mini*) | 37.72 | 26.44 | 61.07 | 58.20 | 45.33 | 36.46 | 45.26 | 36.41 |
| + Upscale (*SplashNet*) | 33.57 | 20.59 | 60.16 | 56.95 | 44.79 | 35.49 | 44.78 | 35.67 |

**Zero-shot generalization.** Table 3 shows that, in the zero-shot regime (i.e. on non-training domains), the joint-hand architecture with RTN and ACM often surpasses the Split-and-Share architecture under greedy decoding, even though the ranking reverses once beam search is applied. The most plausible explanation is that the joint-hand encoder can observe both wrists simultaneously and therefore learns a stronger *implicit* LM, capturing bi- and tri-gram dependencies spread across hands. While that emergent linguistic prior is helpful when no external LM is available, it becomes redundant—and potentially counter-productive—once a more reliable 6-gram LM is introduced at inference time.

Table 4: Finetuned CER (%) with and without beam search.

| Method | Test domain val (greedy) | Test domain val (beam) | Test domain test (greedy) | Test domain test (beam) |
|---|---|---|---|---|
| Sivakumar et al. 2024 | 11.39 | 8.31 | 11.28 | 6.95 |
| + RSG | 12.11 | 6.70 | 12.50 | 6.92 |
| + RTN | **11.83** | 6.47 | **11.75** | 6.63 |
| + ACM | 14.24 | 7.18 | 14.74 | 7.45 |
| + RTN + ACM | 12.70 | 6.55 | 12.90 | 6.53 |
| + Split | 12.67 | 6.04 | 12.89 | 6.10 |
| + Share | 13.01 | 6.13 | 13.22 | 5.87 |
| + FT Unshare | 12.68 | 5.85 | 13.07 | 5.96 |
| + Upscale (SplashNet) | 12.10 | **5.47** | 12.39 | **5.51** |
| + FT Unshare | 11.80 | 5.57 | 12.13 | 5.67 |

**Fine-tuned models.** A different pattern emerges after user-specific fine-tuning (Table 4). None of our variants fully match the baseline of Sivakumar et al. (14) under greedy decoding, despite decisive gains once beam search is enabled. Because the only changes between the baseline and the "+ RSG" model are (i) four-fold longer clips during training, (ii) coarser spectral resolution, and (iii) a five-fold higher probability of channel masking, the degradation must stem from one (or a combination) of these factors.

Two clues implicate aggressive channel masking (ACM). First, the model with RTN but not ACM achieves the best greedy scores, whereas the model with just ACM but not RTN yields the *worst* greedy performance. Second, the performance gap between the model with just RTN and the model with both RTN and ACM vanishes under beam search, indicating that the external LM compensates for information lost when ACM forces the network to rely on low-order feature combinations. We therefore hypothesize that ACM, while beneficial for cross-user generalization, removes within-user cues that help distinguish confusable keystrokes, a weakness that beam-search decoding can largely recover.

To confirm that the implicitly increased channel masking in the "+ RSG" model (and all other models) is largely responsible for the uniformly worse greedy decoding performance we see compared to the baseline of Sivakumar et al. (14), we ran an additional experiment in which we trained a model similar to our +RSG model, but apply SpecAugment on the 33-bin spectrograms from each channel before the bin aggregation of RSG (rather than after). This keeps the extent of masking equivalent to that of Sivakumar et al. 2024 while also allowing us to use the RSG frontend. This model does not perform significantly worse than the baseline of Sivakumar et al. 2024 with greedy decoding, confirming our suspicion that the core reason for the worse greedy decoding results in the finetuned case is increased channel masking.

Table 5: Finetuned CER (%) under greedy decoding. Values are mean ± standard deviation.

| Method | Test domain val (greedy) | Test domain test (greedy) |
|---|---|---|
| Sivakumar 2024 | $11.39 \pm 4.28$ | $11.28 \pm 4.45$ |
| + RSG | $12.11 \pm 4.67$ | $12.50 \pm 4.97$ |
| + RSG w/ pre-aggregation masking | $11.02 \pm 4.32$ | $11.53 \pm 4.70$ |

In summary, greedy decoding accentuates two complementary inductive biases: (1) joint-hand encoders learn a useful internal LM, an advantage that vanishes once an external LM is applied, and (2) aggressive masking of input channels trades cross-user generalization for within-user discriminability, a trade-off that RTN and beam-search decoding can effectively offset.

## A.4 Calculation of FLOPs

FLOPs were measured using FlopTensorDispatchMode in PyTorch with an arbitrary 30-second input.

## A.5 Additional Ablations on ACM Masking, RTN Sliding Windows, and RSG

Table 6: Zero-shot CER (%, mean ± s.d. across participants) for ablations on ACM mask width and RTN sliding window. "-RSG" corresponds to applying ACM on the full-resolution spectrogram as described in the text.

| Method | Other domain val | Test domain val | Test domain test |
|---|---|---|---|
| SplashNet-Mini | $58.20 \pm 10.50$ | $36.46 \pm 7.09$ | $36.41 \pm 7.30$ |
| + 4s SW inference | $60.43 \pm 7.86$ | $37.57 \pm 7.18$ | $37.71 \pm 7.47$ |
| + 4s SW train + inference | $60.43 \pm 7.70$ | $36.95 \pm 7.89$ | $36.69 \pm 7.80$ |
| + 16s SW inference | $57.24 \pm 11.00$ | $36.43 \pm 7.30$ | $36.73 \pm 7.40$ |
| + mask width = 8 | $58.12 \pm 11.72$ | $36.57 \pm 7.02$ | $36.05 \pm 6.74$ |
| + mask width = 16 | $57.65 \pm 10.47$ | $36.66 \pm 7.21$ | $36.92 \pm 7.40$ |
| - RSG | $60.70 \pm 11.23$ | $37.90 \pm 6.58$ | $37.19 \pm 6.33$ |
| Joint-Hand baseline (+ RSG + RTN + ACM) | $58.85 \pm 10.50$ | $36.41 \pm 7.21$ | $36.42 \pm 7.11$ |
| + mask width = 8 | $59.41 \pm 10.63$ | $37.55 \pm 6.97$ | $37.30 \pm 6.90$ |
| + mask width = 16 | $57.17 \pm 11.32$ | $36.95 \pm 7.62$ | $36.47 \pm 7.96$ |

Table 7: Finetuned CER (%, mean ± s.d. across participants) for ablations on ACM mask width and RTN sliding window.

| Method | Test domain val | Test domain test |
|---|---|---|
| SplashNet-Mini | $6.13 \pm 2.96$ | $5.87 \pm 3.04$ |
| + 4s SW inference | $6.10 \pm 2.91$ | $6.28 \pm 3.09$ |
| + 4s SW train + inference | $6.34 \pm 2.86$ | $6.37 \pm 3.10$ |
| + 16s SW inference | $5.76 \pm 2.64$ | $5.74 \pm 2.86$ |
| + mask width = 8 | $5.85 \pm 2.72$ | $6.02 \pm 3.11$ |
| + mask width = 16 | $6.09 \pm 2.88$ | $5.72 \pm 2.96$ |
| - RSG | $5.87 \pm 2.73$ | $5.75 \pm 3.06$ |
| Joint-Hand baseline (+ RSG + RTN + ACM) | $6.55 \pm 2.91$ | $6.53 \pm 3.27$ |
| + mask width = 8 | $6.14 \pm 2.85$ | $6.56 \pm 3.60$ |
| + mask width = 16 | $6.55 \pm 3.09$ | $6.45 \pm 3.18$ |

We conducted additional ablations to examine the sensitivity of model performance to the strength of ACM, the presence of RSG, and the temporal window used for RTN normalization. Unless otherwise noted, none of the differences reported here reached significance under a two-tailed paired $t$-test across participants.

**ACM mask width.**  In our main experiments, ACM uses a maximum frequency mask width of 12 bins, while models without ACM use a maximum width of 4. Here, we trained *SplashNet-mini* and the *Joint-Hand* baseline with maximum mask widths of 8 and 16. As shown in Tables 6 and 7, these variations produced small, nonsignificant changes in CER in both the zero-shot and finetuned settings, indicating that performance is not particularly sensitive to the precise mask width within this range.

**-RSG.**  We also evaluated a variant without RSG, in which ACM was applied to the full-resolution spectrogram by masking a 6-bin dummy vector and inverting the 33-to-6 bin mapping used in RSG to obtain a 33-bin mask. Although performance differences relative to the standard RSG configuration were small and nonsignificant, this variant considerably increases both compute and memory requirements since it has more than sixfold greater input feature dimensionality.

**RTN sliding window.**  Our original RTN normalization uses all past time points within a sample to compute normalization statistics. We additionally evaluated inference-time sliding-window variants with 4-second and 16-second windows. Using a 4-second window led to a small but statistically significant degradation in zero-shot performance ($p < 0.05$, two-tailed), while a 16-second window yielded performance comparable to the default setting. This likely reflects the fact that models were trained on 16-second samples and benefited from matching normalization context at test time. When models were also trained with a 4-second RTN sliding window, this degradation disappeared, and differences were no longer significant in either the zero-shot or finetuned settings.

**Summary.**  Across all tested configurations, performance differences were small and generally nonsignificant. ACM is robust to maximum frequency mask width changes in the range of 8–16 bins. Removing RSG does not significantly degrade performance but incurs higher compute and memory costs. RTN performance remains stable across temporal windows when training and inference are matched, with only a modest increase in CER when using 4 second windows at inference time alone.

## A.6 Aggressive channel masking with mean imputation

Table 8: CER (%, mean ± s.d. across participants) for different ACM masking configurations.

| Method | Test domain val | Test domain test |
|---|---|---|
| Sivakumar et al. 2024 | $52.10 \pm 5.54$ | $51.78 \pm 4.61$ |
| + RSG | $47.26 \pm 5.26$ | $47.18 \pm 5.19$ |
| + ACM | $42.62 \pm 7.18$ | $42.62 \pm 7.10$ |
| + ACM (mean impute) | $47.59 \pm 7.16$ | $48.43 \pm 6.24$ |
| + RTN + ACM | $36.41 \pm 7.21$ | $36.42 \pm 7.11$ |

We performed an ablation to test whether RTN might interact with ACM by stabilizing input features such that the default masking value of 0 corresponds to their per-session mean. When using ACM with RTN, the default mask value of 0 corresponds to the per-sample mean for each feature, whereas under BatchNorm this same value can be far out of distribution for a given feature from a given sample. To test whether it is important that the masking value is the per-sample feature mean, we replaced RTN with standard BatchNorm while setting the ACM masking value to the per-sample mean of each feature, training a Joint-Hand model with RSG, BatchNorm, and ACM using per-channel sample mean imputation.

Somewhat surprisingly, this variant performed on par with the model with RSG alone and substantially worse than the RSG + BatchNorm + ACM configuration with standard zero imputation. This suggests that simply matching the mask value to the per-session mean is not sufficient to reproduce the performance benefits obtained with RTN + ACM, let alone to obtain the benefits of ACM in the absence of RTN. More generally, these results indicate that the choice of masking value being stable across samples (rather than corresponding to the per-sample mean) appears to be the more important factor for ACM's effectiveness in this setting.

## A.7 UMAP analyses on early intermediate representations

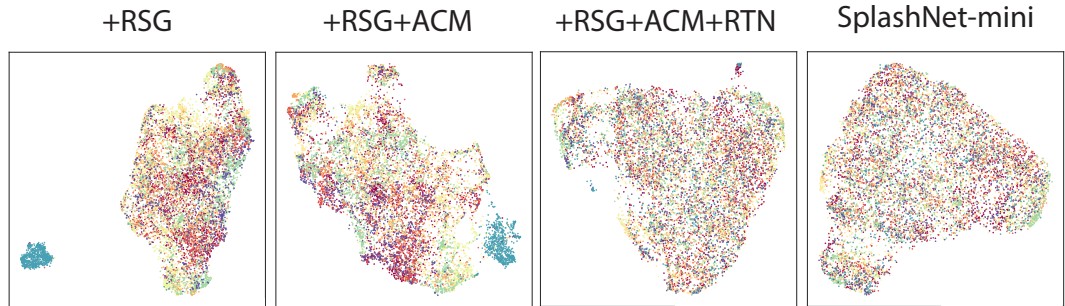

Figure 5: UMAP visualization of model activations after the first TDSConv block for four models (+RSG, +RSG+ACM, +RSG+ACM+RTN, and SplashNet-mini). We extracted activations from every 100th timestep from one session of each of the 8 held-out users. Colors indicate the user identity of each point. In the models without RTN, some users' representations occupy largely disjoint regions of the activation manifold, whereas models with RTN (+RSG+ACM+RTN and SplashNet-mini) produce markedly more overlapping per-user representations, indicating improved cross-user alignment in the learned feature space.

## A.8 Alternative train-test split analyses

Table 9: CER (%, mean ± s.d. across participants) with a new 78/18 train/validation split. The first column shows performance on the *same 18 users* when they were included in the original training set (i.e., the previous training domain validation performance), while the remaining columns show results when these 18 users are held out for validation. The test set remains the same 8 held-out users.

| Method | Train domain val (prev. split) | Train domain val (18 held-out) | Test domain val (18 held-out) | Test domain test (18 held-out) |
|---|---|---|---|---|
| + RSG | 13.52 ± 7.59 | 58.42 ± 10.43 | 47.89 ± 6.36 | 49.01 ± 5.35 |
| + RTN | 13.09 ± 6.32 | 53.99 ± 11.14 | 42.81 ± 6.75 | 42.62 ± 6.89 |
| + ACM | 23.47 ± 9.74 | 55.95 ± 10.46 | 44.59 ± 6.47 | 45.10 ± 7.26 |
| + RTN + ACM | 21.71 ± 9.67 | 51.25 ± 11.49 | 38.72 ± 6.63 | 39.05 ± 7.22 |
| + Split | 23.93 ± 10.99 | 51.57 ± 11.78 | 39.81 ± 6.72 | 40.14 ± 7.16 |
| + Share | 26.44 ± 10.64 | 49.84 ± 12.57 | 38.60 ± 7.52 | 38.45 ± 7.86 |

## A.9 Analyses on emg2pose dataset

Table 10: Mean angular error (degrees) and landmark distance (mm) under different generalization regimes. All results are in the "tracking" setting.

| Generalization | Model | Angular Error | Landmark Dist |
|---|---|---|---|
| User | RTN+ACM | 7.5951 | 10.1567 |
| Stage | RTN+ACM | 11.1243 | 15.2404 |
| User, Stage | RTN+ACM | 10.8433 | 15.3644 |
| User | RTN | 7.6007 | 10.1641 |
| Stage | RTN | 11.1627 | 15.2241 |
| User, Stage | RTN | 10.9085 | 15.3851 |
| User | ACM | 7.6388 | 10.2394 |
| Stage | ACM | 11.2601 | 15.4489 |
| User, Stage | ACM | 11.0343 | 15.6413 |
| User | Baseline | 7.6549 | 10.2585 |
| Stage | Baseline | 11.2892 | 15.4360 |
| User, Stage | Baseline | 11.1222 | 15.7388 |

Although the focus of this work is on EMG keystroke decoding, we also performed preliminary experiments to assess how RTN and ACM transfer to the `emg2pose` benchmark (13). A key consideration is that the baseline model of Salter et al. (13) uses a learned convolutional featurizer that immediately mixes signals across all electrodes, rather than spectrogram-based per-electrode features as in our keystroke decoding setting. This architectural difference makes a direct application of RTN and ACM less straightforward.

To adapt these methods, we applied RTN and ACM to the outputs of the first convolutional layer in the featurizer. Since these intermediate features are not spectrograms, ACM was implemented by randomly zeroing out 50% of the features in one-third of training samples and 75% in another third. The resulting performance, summarized in Table 10, shows minimal differences relative to the baseline in both mean angular error and landmark distance across user, stage, and combined generalization regimes.

These findings are somewhat unsurprising: applying ACM after early feature mixing likely limits its ability to enforce robust low-order structure, and RTN is less meaningful without clear per-electrode feature boundaries. Future work should explore applying these methods directly to the raw EMG signal before featurization, or using architectures with explicit per-electrode feature streams (e.g., spectrogram-based or structured learned featurizers), where RTN and ACM may have stronger effects.

**A.10   Compute Resources**

All experiments were run on a single RTX 4090 (with 24GB VRAM) GPU. Each generic model took roughly 24 hours to train. Finetuning took 20-30 minutes for each of the 8 test users, and an additional 30 minutes to evaluate with beam search for each of the 8 test users.

**A.11   Dataset License**

The emg2qwerty dataset (14) is available under the CC-BY-NC-4.0 license, which this work is compliant with.

