# OpenReview forum: "SplashNet: Split‑and‑Share Encoders for Accurate and Efficient Typing with Surface Electromyography"
_NeurIPS.cc/2025/Conference — NeurIPS 2025 poster_

### Official Review · Reviewer_zN88 · 2025-06-13

**Clarity:** 2
**Significance:** 3
**Originality:** 3
**Rating:** 5
**Confidence:** 4

**Summary:**

This paper addresses the poor cross-user generalization and high error rates in surface electromyography based typing systems. The authors identify three primary issues in the existing emg2qwerty baseline: mismatched signal statistics across users, a fragile dependency on high-order feature combinations, and the lack of an architectural design that reflects the bilateral nature of typing. To solve these problems, the paper introduces three modifications: Rolling Time Normalization to adaptively align input signal distributions; Aggressive Channel Masking, a data augmentation technique that encourages the model to rely on more robust and generalizable low-order features ; and a Split-and-Share encoder architecture that processes signals from each hand independently with weight-shared streams. The resulting model, SplashNet, establishes a new state-of-the-art, reducing the zero-shot character error rate by a relative 31% to 35.7% and the fine-tuned CER by a relative 21% to 5.5%. Notably, these improvements are achieved with greater efficiency, as the compact SplashNet-mini model uses only one-quarter of the parameters and 60% of the FLOPs of the baseline.

**Questions:**

The paper attributes its success to specific signal processing choices, including adopted frequency bands and a powerful normalization method. However, since these were evaluated on a validation set described as "notably noisier," it is necessary to clarify whether the performance gains reflect a fundamentally better approach or if they are an artifact of the proposed methods being uniquely effective on this non-standard, noisy data.

**Ethical Concerns:**

["NO or VERY MINOR ethics concerns only"]

**Final Justification:**

The authors have addressed most of my concerns. Therefore, I decided to raise my score to 5.

**Limitations:**

**Limitations**
1. The state-of-the-art results reported in the main text are achieved using beam-search decoding with an external 6-gram language model. The appendix, however, shows that under greedy decoding (without a language model), the performance of the proposed models can be inferior to the baseline, especially after fine-tuning. This suggests that the core model's intrinsic ability to discriminate signals may be reduced, and the system depends heavily on the language model to achieve its top performance.
2. While the authors' critique of the original validation set is valid, their proposed "other domain validation set" is built from only four participants. The paper also states these participants exhibit "notably noisier EMG data". A validation set that is both small and potentially unrepresentative in its data quality may not provide a stable foundation for model selection, raising questions about the generalizability of the reported validation performance.
3. The paper concludes that unsharing the encoder weights during the fine-tuning phase offers no performance benefit, suggesting that hand-specific adapted weights are unnecessary. This conclusion is drawn from experiments with a limited amount of user-specific data. It is plausible that with more extensive fine-tuning data, allowing the model to adapt to subtle neuromuscular asymmetries between a user's left and right hands could yield further gains. Thus, the claim that this strategy is broadly unnecessary may be an overstatement based on the current evidence.

**Paper Formatting Concerns:**

No Paper Formatting Concerns

**Quality:**

3

**Strengths And Weaknesses:**

**Strengths**
1. The research begins by conducting a precise diagnosis of the specific weaknesses in the prior state-of-the-art model. This analytical approach ensures that the proposed solutions are well-motivated and directly target the root causes of poor performance.
2. The proposed modifications are computationally inexpensive and grounded in domain knowledge of signal processing and biomechanics. The Split-and-Share architecture, in particular, effectively incorporates an inductive bias for the bilateral symmetry of typing , which not only improves accuracy but also significantly reduces the model's parameter count and computational complexity.

---

> ### Author Rebuttal · Authors · 2025-07-30
>
> We appreciate Reviewer zN88's praise for our targeted diagnosis of baseline weaknesses and efficient, biomechanically grounded solutions, alongside their valid concerns about validation stability and decoding dependencies. Our rebuttal clarifies the generalizability of our gains and commits to new dataset splits and expanded discussion of tradeoffs.
>
> **Q1:** *The paper attributes its success to specific signal processing choices, including adopted frequency bands and a powerful normalization method. However, since these were evaluated on a validation set described as "notably noisier," it is necessary to clarify whether the performance gains reflect a fundamentally better approach or if they are an artifact of the proposed methods being uniquely effective on this non-standard, noisy data.*
>
> - Thank you for raising this point. We wish to clarify that the state-of-the-art performance we achieve for all 8 held-out test users (Figure 4), all of whom have standard-quality data, is strong evidence that our performance gains reflect a fundamentally better approach that is not “uniquely effective on this non-standard, noisy data”. In principle, it is possible that using the other domain validation set might select for models that retain higher performance for the 4 noisy validation users, potentially at the cost of performance on the cleaner 8 held-out test users. However, we empirically observe that performance on the other domain validation set largely tracks with performance on the test set users. Nonetheless, we will run experiments with a new train/val/test split to address concerns regarding the other domain validation set, which we address further in L2.
>
> **L1:** *The state-of-the-art results reported in the main text are achieved using beam-search decoding with an external 6-gram language model. The appendix, however, shows that under greedy decoding (without a language model), the performance of the proposed models can be inferior to the baseline, especially after fine-tuning. This suggests that the core model's intrinsic ability to discriminate signals may be reduced, and the system depends heavily on the language model to achieve its top performance.*
>
> - We agree that this is an important limitation of our work, and though we address it as such in our discussion (and further in the appendix), we will expand on the trade-offs involved in the results and discussion sections of our revised draft, as we believe it provides important insights into how our models work. As emphasized in Sections 5.1,  5.2, and Appendix  A.3, our primary evaluation metric is beam-search decoding with a 6-gram character LM, which we view as the most realistic configuration for practical deployment. This mirrors standard practice in CTC-based speech pipelines, where an external LM adds negligible compute yet is essential for disambiguating confusable outputs in noisy, real-time settings such as EMG-based typing. Greedy decoding, while simpler, is not representative of assistive-technology or AR/VR interfaces, which can accommodate a small-beam search without violating latency budgets but still demand high accuracy. In precisely this realistic setup our Split-and-Share models achieve state-of-the-art performance, yielding relative CER reductions of 31% in the zero-shot regime and 21% after fine-tuning versus the baseline of Sivakumar et al. 2024.
> - We note two settings where beam search and greedy decoding results diverge. First, when no LM is used, the Split-and-Share models generally perform worse than the Joint-Hand models. However, when an LM is used, the Split-and-Share models generally outperform the Joint-Hand models. Please note the same LM is used for all models. These results therefore support the view that the Split-and-Share models better discriminate aspects of the EMG signal that are complementary to an LM prior, whereas the Joint-Hand models better discriminate aspects of the EMG signal which are redundant to an LM prior. Importantly, this interpretation is consistent with the inductive biases allotted by the two architectures: Co-articulatory activations between hands processed by the Joint-Hand model are likely to correlate with n-grams in the text, and hence are somewhat redundant with an n-gram LM. The Split-and-Share model cannot learn such co-articulatory patterns, and hence must learn other aspects of the EMG signal, such as the within-hand mapping from EMG signals to keystroke movements, which in turn are less redundant with an n-gram LM. We will incorporate this into the discussion.
> - The second setting where beam search and greedy decoding results diverge is in the fine-tuned case, where we observe that increased channel masking causes a decrease in fine-tuned model performance without an LM. Because all of our models have reduced spectral granularity, and reduced spectral granularity implicitly increases channel masking even before ACM, we observe that all of our models perform worse than the baseline from Sivakumar et al. 2024 in the fine-tuned case when using greedy decoding. Though we speculated in the previous draft that this was due to the increased channel masking (rather than the reduced spectral granularity itself), we now provide an experiment confirming this. Specifically, we trained a model similar to our +RSG model, but apply SpecAugment on the 33-bin spectrograms from each channel before the bin aggregation of RSG (rather than after). This keeps the extent of masking equivalent to that of Sivakumar et al. 2024 while also allowing us to use the RSG frontend. This model does not perform significantly worse than the baseline of Sivakumar et al. 2024 with greedy decoding, confirming our suspicion that the core reason for the worse greedy decoding results in the finetuned case is increased channel masking. This aligns with our statement that “aggressive masking of input channels trades cross-user generalization for within-user discriminability, a trade-off that RTN and beam-search decoding can effectively offset.” We will further highlight these points in the main text of our revised manuscript.
> |  | Test domain val (greedy) | Test domain test (greedy)|
> | --- | --- | --- |
> | Sivakumar 2024 | 11.39 ± 4.28 | 11.28 ± 4.45 |
> | + RSG | 12.11 ± 4.67 | 12.50 ± 4.97 |
> | + RSG w/ pre-aggregation masking | 11.02 ± 4.32 | 11.53 ± 4.70 |
> - In summary, our models' design choices enhance performance under practical conditions, and we appreciate this feedback for deepening our analysis.
>
> **L2:** *While the authors' critique of the original validation set is valid, their proposed "other domain validation set" is built from only four participants. The paper also states these participants exhibit "notably noisier EMG data". A validation set that is both small and potentially unrepresentative in its data quality may not provide a stable foundation for model selection, raising questions about the generalizability of the reported validation performance.*
>
> - The generalizability of the reported validation performance is evident in the state-of-the-art results we achieve on the 8 held-out test users with standard-quality data. Nonetheless, we agree that using only 4 users with noisy data as a validation set is far from ideal, regardless of its superiority to a training domain validation set. We made this choice as it was the only option that allowed us to have a held-out user validation set while also maintaining the official train and test sets from Sivakumar et al. 2024, making it easier to compare our results to theirs. However, to address concerns regarding the other domain validation set, we will repeat our main experiments using a new dataset split, where we will use 80% of the official training users for training, 20% of these users for validation, and the same 8 users as Sivakumar et al. 2024 for testing (the “other domain validation set” users will be excluded completely as in Sivakumar et al. 2024). Given the large and consistent improvements our methods show compared to Sivakumar et al. 2024 on all held-out test users (Figure 4), we believe the present results are robust, and we do not anticipate any noteworthy differences when using a new split.
>
> **L3:** *The paper concludes that unsharing the encoder weights during the fine-tuning phase offers no performance benefit, suggesting that hand-specific adapted weights are unnecessary. This conclusion is drawn from experiments with a limited amount of user-specific data. It is plausible that with more extensive fine-tuning data, allowing the model to adapt to subtle neuromuscular asymmetries between a user's left and right hands could yield further gains. Thus, the claim that this strategy is broadly unnecessary may be an overstatement based on the current evidence.*
>
> - We agree that per-hand weights might become advantageous with more user-specific data, and will make this explicit in our revised draft.

---

> > ### Comment · Reviewer_zN88 · 2025-08-06
> > **Thank you for your detailed rebuttal**
> >
> > I appreciate the authors' detailed response. I have read the author's response, which addresses most of my concerns. After careful consideration and taking the other reviews into account, I have decided to raise my score to 5.

---

### Official Review · Reviewer_zecG · 2025-06-29

**Clarity:** 4
**Significance:** 3
**Originality:** 3
**Rating:** 5
**Confidence:** 5

**Summary:**

The paper proposes three simple but effective improvements on emg2qwerty that demonstrate a better zero-shot and fine-tuning performance on sEMG to typing decodings. The proposed three improvements are well motivated, including the per-feature normalisation across time, aggressive channel masking for data augmentation, and shared weights when processing signals from left and right hands. With the three improvements, the model substantially reduces the character error rate for both zero-shot and fine-tuning tasks and reduces computational costs as well. This demonstrates its potential for advancing EMG to typing neuromotor interfaces in AR/VR applications.

**Questions:**

1. As shown in table one and table two, the three major contributions are tested on top of the reduced spectral granularity. Wondering the performance if keeping the high spectral resolution but applying aggressive channel masking. Will the fine-grained aggressive masking produce different results?
2. Is it possible to keep both low-order features that are more generalisable across subjects as well as high-order features that are more reliable during each subject’s fine tuning, e.g., by using learnable masks/frequency features?
3. I wondered what the feature space looks like after applying the three improvements. Are the features more overlapped across subjects when visualising using clustering methods/t-SNE/UMAP?
4. Nit: line 78: misspelling of aggressive

**Ethical Concerns:**

["NO or VERY MINOR ethics concerns only"]

**Final Justification:**

Thanks for the response. My critiques have been well addressed. Looking forward to the analysis and the future work.

**Limitations:**

As the authors mentioned in the discussion, the proposed method has only been validated on data from healthy subjects. The other point the authors might want to discuss is the potential applications in cross-session performance especially when the band is put on and off with electrode shifts.

**Paper Formatting Concerns:**

No observed formatting issues.

**Quality:**

4

**Strengths And Weaknesses:**

Strengths
1. The three key contributions of this work are reasonably proposed given observations and learnings from the sota model. All of them are solid with clear motivations.
2. The ideas are well presented.
3. The authors did comprehensive experiments to validate the effectiveness of the three contributions.

Weaknesses

Barely found any weaknesses from the perspective of soundness, presentation, and contribution. Please see the open questions below.

---

> ### Author Rebuttal · Authors · 2025-07-30
>
> We thank Reviewer zecG for their enthusiastic feedback on our solid motivations, clear presentation, and comprehensive experiments, as well as their thoughtful questions on spectral granularity, learnable masking, and feature visualization. In our rebuttal, we address these by proposing learnable masks as a future extension, while committing to add clustering visualizations and masking without reduced spectral granularity in the revised draft.
>
> **Q1:** *As shown in table one and table two, the three major contributions are tested on top of the reduced spectral granularity. Wondering the performance if keeping the high spectral resolution but applying aggressive channel masking. Will the fine-grained aggressive masking produce different results?*
>
> - We switched to RSG early on because the frequency bins used in the original model were considerably more granular than is standard for sEMG. For example, CTRL-Labs et al. 2024 used features in the same six frequency bands that we use with RSG. When we saw that performance didn’t decrease and in fact slightly improved, we decided to keep this change for the rest of our experiments, as it also reduced training time, parameter count, and FLOPs. It is possible that ACM might produce different results without RSG, and we will include results with ACM without RSG in the revised draft. Nonetheless, we generally believe RSG is an improvement over the highly granular spectrogram of Sivakumar et al. 2024.
>
> **Q2:** *Is it possible to keep both low-order features that are more generalisable across subjects as well as high-order features that are more reliable during each subject’s fine tuning, e.g., by using learnable masks/frequency features?*
>
> - It might be possible to learn such feature masks, and this is an interesting direction for future development. We will include this as a future direction in our discussion.
>
> **Q3:** *I wondered what the feature space looks like after applying the three improvements. Are the features more overlapped across subjects when visualising using clustering methods/t-SNE/UMAP?*
>
> - We think this could make for an insightful analysis, similar to what Ctrl-Labs et al. 2024 provided in Fig 4 of their work, and we would be happy to include such visualizations in our revised draft.
>
> **Q4:** *Nit: line 78: misspelling of aggressive*
>
> - Thank you for pointing this out. We have now corrected this spelling error.
>
> **L1:** *As the authors mentioned in the discussion, the proposed method has only been validated on data from healthy subjects. The other point the authors might want to discuss is the potential applications in cross-session performance especially when the band is put on and off with electrode shifts.*
>
> - We have now included a more explicit discussion of what our approach means for cross-session performance. In particular, the emg2qwerty dataset examines cross-session performance, both in the zero-shot and finetuned settings. That is, the data used for validation and testing all comes from different sessions (and in the zero-shot setting, new users), between which the participants doffed and donned the EMG wristbands (as per the description in the emg2qwerty paper). Hence, our results directly reflect how models might perform in new sessions when the band is put on and off.

---

> > ### Comment · Reviewer_zecG · 2025-08-02
> >
> > Thanks for the response. My critiques have been well addressed. Looking forward to the analysis and the future work.

---

### Official Review · Reviewer_zDKU · 2025-06-30

**Clarity:** 3
**Significance:** 2
**Originality:** 3
**Rating:** 4
**Confidence:** 3

**Summary:**

This paper introduces SplashNet, a novel architecture for surface electromyography (sEMG)-based keystroke decoding. The authors address key challenges in cross-user generalization by proposing three innovations: (1) ​​Rolling Time Normalization (RTN)​​ for adaptive per-session signal standardization, (2) ​​Aggressive Channel Masking (ACM)​​ to encourage robust low-order feature dependencies, and (3) a ​​Split-and-Share encoder​​ leveraging bilateral symmetry of typing biomechanics. Evaluated on the emg2qwerty benchmark, SplashNet-mini reduces character error rates (CER) to 36.4% zero-shot and 5.9% after fine-tuning while using 1/4 the parameters and 60% FLOPs of prior work.

**Questions:**

1. Why were the 4 low-session users selected for "other domain validation"? Could this choice favor models overfit to noisy data? Providing results on the official validation set alongside Table 1 would clarify generalization.

​​2. Does RTNs online normalization enhance ACMs feature-subset regularization? Please provide an analysis like mutual information between masked features which could disentangle their interaction.

3. The paper assumes that "low order feature similarity" is the direct reason for ACM to improve generalization, but lacks cross user similarity experiments on feature subsets

**Ethical Concerns:**

["NO or VERY MINOR ethics concerns only"]

**Final Justification:**

I appreciate the authors' comprehensive response, which has adequately addressed several of my initial concerns. In light of these improvements, I have decided to revise my score upward.

**Limitations:**

The authors partially address limitations. However, there is a lack of cross user similarity experiments on feature subsets, and sensitivity analysis on masking ratios is not provided

**Paper Formatting Concerns:**

None. The paper adheres to NeurIPS formatting guidelines with clear sections, equations, and tables.

**Quality:**

2

**Strengths And Weaknesses:**

1. Well-structured with clear motivation.This article proposes persuasive improvements based on empirical findings of signals.
2. Rigorous evaluation via ablation studies validates each component’s contribution. The 5.5% fine-tuned CER significantly advances state-of-the-art
3.The bilateral Split-and-Share design is a biologically inspired novelty, while ACM rethinks augmentation for EMG-specific challenges.
Weaknesses:
1. ​​Validation Set Justification​​: This method lacks validation on other datasets.
2.The paper did not provide a sensitivity analysis of masking ratios, and zero sample generalization experiments were conducted with different masking ratios (30%/50%/70%)

---

> ### Author Rebuttal · Authors · 2025-07-30
>
> We are grateful to Reviewer zDKU for highlighting our well-motivated improvements and rigorous ablations, while pointing out areas for deeper analysis like validation sets and feature interactions. Our rebuttal includes new sensitivity analyses on masking ratios, training domain validation set results, clarifications on RTN-ACM synergies, and reframing of cross-user feature similarity as a hypothesis.
>
> **W1:** *Validation Set Justification: This method lacks validation on other datasets.*
>
> - For this comment, we were unsure if you were referring to concerns with our validation set or were pointing out that we only used one dataset (emg2qwerty). We address the latter point here, and we thoroughly address the former point in our response to Q1.
> - We agree that external validation is important. However, among publicly available datasets, emg2qwerty is the only resource that simultaneously offers naturalistic, continuous touch typing and the user/session scale required to study zero‑shot generalization and model personalization. Other public sEMG typing datasets either comprise isolated, metronome‑paced keypresses with small cohorts (19 users, two sessions) [1] or target password‑specific security settings (37 participants) [2], which do not reflect naturalistic typing or provide sufficient diversity for large‑scale cross‑user generalization experiments.
>
> [1] Eby et al., Electromyographic typing gesture classification dataset for neurotechnological human-machine interfaces.
>
> [2] Gazzari et al., My(o) Armband Leaks Passwords: An EMG and IMU Based Keylogging Side-Channel Attack
>
> **W2:** *The paper did not provide a sensitivity analysis of masking ratios, and zero sample generalization experiments were conducted with different masking ratios (30%/50%/70%)*
>
> - We have now performed sensitivity analyses for the masking ratios and will add further sweeps to the final paper. We would like to clarify that zero-shot generalization experiments were not conducted with different masking ratios unless otherwise noted (e.g. models with ACM have a different masking ratio than those without it, but otherwise all of our models have the same masking ratios).
> - For our new experiments, we trained SplashNet-mini models with maximum frequency mask width set to 8 and 16 (the default value with ACM was 12, and the default value without ACM was 4). We observe nonsignificant differences in CER, indicating that the model is not particularly dependent on the strength of ACM within this range. We will further sweep mask width on all of our main models (RSG+RTN+ACM, SplashNet-mini, SplashNet) for the final manuscript.
> - Results in the zero-shot setting:
> |  | Other domain val | Test domain val | Test domain test |
> | --- | --- | --- | --- |
> |SplashNet-Mini|58.20 ± 10.50| 36.46 ± 7.09| 36.41 ± 7.30|
> |+ mask width = 8|58.12 ± 11.72| 36.57 ± 7.02| 36.05 ± 6.74|
> |+ mask width = 16|57.65 ± 10.47| 36.66 ± 7.21| 36.92 ± 7.4|
> - Results in the finetuned setting:
> |  | Test domain val | Test domain test |
> | --- | --- | --- |
> | SplashNet-Mini|6.13 ± 2.96|5.87 ± 3.04|
> |+ mask width = 8|5.85 ± 2.72|6.02 ± 3.11|
> |+ mask width = 16|6.09 ± 2.88|5.72 ± 2.96|
>
> **Q1:** *Why were the 4 low-session users selected for "other domain validation"? Could this choice favor models overfit to noisy data? Providing results on the official validation set alongside Table 1 would clarify generalization.*
>
> - We would like to clarify some potential misunderstandings. Neither us nor Sivakumar et al. 2024 use these 4 users for model training: Sivakumar et al. 2024 doesn’t use them at all, whereas we use them as the “other domain validation set”. Hence, our models cannot “overfit” to these particular users in the sense of learning user-specific strategies during training. It is possible that the other domain validation set might select for models with higher performance for these noisy held-out users, potentially at the cost of performance on the cleaner 8 test users. However, we empirically observe that performance on the other domain validation set mostly tracks with performance on the test set users (Table 1).
> - The same cannot be said for the training domain validation set recommended by Sivakumar et al. 2024. Though running this evaluation with LM beam search is time intensive (requiring ~20 hours per model due to the non-parallelizability of beam search and the size of the training domain validation set), we will provide these results in the final paper. For now, we provide the results with greedy decoding (no LM) below, alongside the results in our current Table 3. We will include results with beam search in both Table 1 and Table 3 in our revised draft.
> | |Train domain val (greedy)|Other domain val (greedy)|Other domain val (beam)|Test domain val (greedy)|Test domain val (beam)|Test domain test (greedy)|Test domain test (beam)
> |-|-|-|-|-|-|-|-|
> |Sivakumar et al. 2024|22.51|72.44|72.07|55.57|52.10|55.38|51.78
> |+ RSG|23.59|68.55|67.48|52.24|47.26|52.27|47.18
> |&nbsp;+ RTN|23.49|64.19|61.95|46.31|39.49|46.07|39.15
> |&nbsp;+ ACM|34.77|65.60|63.08|48.87|42.62|49.23|42.62
> |&nbsp;+ RTN + ACM|32.85|61.82|58.85|43.72|36.41|43.80|36.42
> |&nbsp;&nbsp;+ Split|35.59|61.74|58.64|45.73|37.28|45.69|37.37
> |&nbsp;&nbsp;&nbsp;+ Share (SplashNet-mini)|37.72|61.07|58.20|45.33|36.46|45.26|36.41
> |&nbsp;&nbsp;&nbsp;&nbsp;+ Upscale (SplashNet)|33.57|60.16|56.95|44.79|35.49|44.78|35.67
> - We can clearly see that the official training domain validation set is remarkably misleading if one’s goal is to transfer to new users. Though our methods show the best performance on the other domain validation set and the test domain sets (i.e. the only sets consisting of held-out users), we get poor performance on the training domain validation set. In particular, RTN and ACM, methods which led to unambiguous, statistically significant gains in zero-shot performance on the held-out user sets, produce either no effect or harm performance according to the training domain validation set.
> - We would also like to note that most other papers on cross-user EMG generalization (e.g., emg2pose and CTRL Labs et al. 2024) use held-out users for validation, whereas emg2qwerty makes the unusual choice to use the same users for both training and validation. Training domain validation performance has little practical value—even Sivakumar et al. 2024 did not report it—since the goal of generic models is transfer to new users, not optimization for trained ones. We are happy to clarify this in the paper.
> - We agree that using only 4 users with noisy data for validation is not ideal, regardless of its superiority to a training domain validation set. We made this choice as it was the only option that allowed us to have a held-out user validation set while also maintaining the official train and test sets from Sivakumar et al. 2024, making it easier to compare our results to theirs. However, to address concerns regarding the other domain validation set, we will repeat our main experiments using a new dataset split, where we will use 80% of the official training users for training, 20% of these users for validation, and the same 8 users as Sivakumar et al. 2024 for testing (the “other domain validation set” users will be excluded completely as in Sivakumar et al. 2024). Given the large and consistent improvements our methods show compared to Sivakumar et al. 2024 on all held-out test users (Figure 4), we believe the present results are robust, and we do not anticipate any noteworthy differences when using a new split.
>
> **Q2:** *Does RTNs online normalization enhance ACMs feature-subset regularization? Please provide an analysis like mutual information between masked features which could disentangle their interaction.*
>
> - RTN’s online normalization likely does work synergistically with ACM. In particular, RTN makes the 0 value used for masking more meaningful: for any participant, it is the mean value that any channel occupies within a given session. On the other hand, when using the default batch normalization of Sivakumar et al. 2024, a value of 0 could be far out of distribution for any given user. We will make this point explicit in our revised draft. We will also include a new ablation in which we use the batch normalization of Sivakumar et al. 2024, but with the masking value set to the mean value occupied by a channel within a given session. In principle, this should disentangle whether the within-user mask value stabilization that RTN induces is itself helpful for improved decoding when separated from the more general within-session normalization provided by RTN.
> - We favor this analysis over a mutual information based analysis, since reliable estimation of high-dimensional mutual information is itself an area of research, and can be significantly affected by hyperparameters including distribution discretization. If our analysis does not satisfy your concern, we are happy to further discuss to be sure we fully address this concern. Thank you!
>
> **Q3:** *The paper assumes that "low order feature similarity" is the direct reason for ACM to improve generalization, but lacks cross user similarity experiments on feature subsets*
>
> - We agree and will revise the manuscript to state low‑order cross‑user feature similarity as a hypothesis, not a demonstrated mechanism. Our scope is improving EMG keystroke decoding, and our empirical result is that ACM consistently improves generalization to unseen users. ACM randomly masks channels/electrodes during training, which discourages reliance on user‑specific, high‑order conjunctions and encourages solutions built from lower‑order feature combinations. We agree that this inductive bias is compatible with, but does not prove, greater cross‑user similarity within feature subsets. We will label this as indirect evidence, temper the language accordingly, and add a limitation noting that validating the mechanism would require further cross-user similarity experiments, which we leave for future work.

---

> > ### Comment · Reviewer_zDKU · 2025-08-05
> > **Rebuttal response**
> >
> > I appreciate the authors' comprehensive response, which has adequately addressed several of my initial concerns. In light of these improvements, I have decided to revise my score upward.

---

### Official Review · Reviewer_5Ux2 · 2025-07-02

**Clarity:** 4
**Significance:** 3
**Originality:** 2
**Rating:** 5
**Confidence:** 4

**Summary:**

This paper revisits the emg2qwerty baseline for decoding typing movements from sEMG. They add several features to the network: (1) a rolling time normalization (2) additional agressive channel-time masking (3) weight sharing across encoders. Overall this reduces parameter counts and improves performance on the baseline by approximately 25-40% and for small networks reduces computation time. They present a set of ablations that expose which features are most important for the improved performance.

**Questions:**

See weaknesses above

**Ethical Concerns:**

["NO or VERY MINOR ethics concerns only"]

**Final Justification:**

Based on inclusion of additional dataset to show generality of the approach I raised my score to a 5, and looking at the feedback it seems there is general interest in the approach.

**Limitations:**

yes

**Quality:**

3

**Strengths And Weaknesses:**

Strengths
* Improves the baseline performance on emg2qwerty
* Paper is well written and easy to follow and makes clear the relevant points
* The ablations are clear and some of the details are interesting, e.g. the other domain validation set


Weaknesses above
* Improvements are somewhat incremental; the novelty of these is more based on a combination of parts rather than any individual modeling innovation, e.g. changing the masking fraction or z-scoring input or sharing weights alone would be quite insufficient. But the paper is very clear and well executed.
 * An experiment on an additional EMG dataset showing the generality of some of these advances (e.g. emg2pose or ninapro) would be beneficial.
* The architectural innovations and experiments could be greater, e.g. exploring lightweight sharing across hands before the character encoding to capture aspects of coarticulation.
* There isn’t quantitative experiments examining different types of rolling time normalization or the amount of masking. These should be further developed and quantified.

---

> ### Author Rebuttal · Authors · 2025-07-30
>
> We appreciate Reviewer 5Ux2's recognition of our improved performance, clear writing, and clear ablation studies, along with their constructive suggestions for expanding experiments on other hyperparameters, architectures and EMG datasets. In our rebuttal, we respond to these by providing new quantitative results on masking and normalization variants, and we discuss hybrid architectures and extensions to other EMG tasks as promising future directions.
>
> **W1:** *Improvements are somewhat incremental; the novelty of these is more based on a combination of parts rather than any individual modeling innovation, e.g. changing the masking fraction or z-scoring input or sharing weights alone would be quite insufficient. But the paper is very clear and well executed.*
>
> - We thank the reviewer for their comments regarding our paper’s clarity and execution. We agree that the key novelty of our work was not to propose model components that are altogether new for machine learning, but rather to derive very simple and inexpensive algorithmic improvements from insights into the data and task structure of EMG keystroke decoding. While we believe research into entirely new methods will be invaluable for further improvements, we also believe that given the early stage of this benchmark, it is appropriate to seek simple and effective solutions that provide insight into the problem. We emphasize that these simple and effective solutions are also more likely to have significant impact on real-time compute-limited devices.
>
> **W2:** *An experiment on an additional EMG dataset showing the generality of some of these advances (e.g. emg2pose or ninapro) would be beneficial.*
>
> - Emg2pose and Ninapro both investigate inherently unimanual tasks, so the Split-and-Share macro-architecture does not apply in these cases. While we believe that the rationale for Rolling Time Normalization and Aggressive Channel Masking should also apply to the emg2pose and Ninapro datasets, the focus of this paper was on the keyboard typing task, and the emg2qwerty dataset is by far the largest and highest quality publicly available dataset for this task. We therefore decide to leave these investigations into other tasks for future work, and will make this clear in the Discussion.
>
> **W3:** *The architectural innovations and experiments could be greater, e.g. exploring lightweight sharing across hands before the character encoding to capture aspects of coarticulation.*
>
> - We appreciate the suggestion and agree that exploring architectures with lightweight joint-hand processing before character decoding could offer a compelling way to combine the strengths of both Split-and-Share (e.g., improved within-hand EMG-to-keystroke modeling) and Joint-Hand (e.g., improved coarticulation modeling) approaches. In this work, we intentionally focused on evaluating the benefits of the Split-and-Share architecture in order to isolate and highlight the advantages of improved within-hand processing. That said, we believe hybrid approaches, such as introducing cross-hand interactions at intermediate layers, are a promising direction for future work. However, we also view this as a substantial and orthogonal design space, deserving of focused investigation in its own right. We will explicitly discuss this in our revised draft as an important direction for future work.
> - We also note that the architectural constraints imposed by the Split-and-Share design bring practical advantages. Aside from parameter and compute efficiency, our architecture is well-suited for future directions such as fully on-wrist typing systems, leveraging unimanual foundation models, or adapting the trained encoders for other unimanual finger-tapping tasks. We will make these points more explicit in our revised draft.
>
> **W4:** *There isn’t quantitative experiments examining different types of rolling time normalization or the amount of masking. These should be further developed and quantified.*
>
> - We fully agree that such results would greatly strengthen our paper. We have swept two additional masking levels for the SplashNet-Mini model, along with two inference-time sliding window variations of rolling time normalization.
> - For our new experiments, we trained SplashNet-mini models with maximum frequency mask width set to 8 and 16 (the default value for ACM in our paper was 12, and the default value without ACM was 4). We observe small, nonsignificant differences in CER, indicating that the model is not particularly dependent on the strength of ACM within this range. We will further sweep frequency mask width on all of our main models (RSG+RTN+ACM, SplashNet-mini, SplashNet) for the final manuscript.
> - We also evaluated inference-time variants of RTN using a sliding window. Though our original RTN used all previous time points in a sample to normalize the data, we have now run experiments with two sliding-window settings: using a 4 second and 16 second sliding window for input feature normalization. We observed a slight degradation of performance when using only a 4 second window but no significant difference when using a 16 second window. However, it is worth noting that the model was trained on samples of length 16 seconds, with RTN using data from up to 16 seconds for normalization during training. In our revised draft, we will further include experiments where we train with RTN using a sliding window of up to 4 seconds during training, to see if this could improve robustness when using a 4 second sliding window at inference. We are happy to include any other types of rolling time normalization that the reviewer would like to see in the revised draft.
> - Results in the zero-shot setting:
> |  | Other domain val | Test domain val | Test domain test |
> | --- | --- | --- | --- |
> | SplashNet-Mini | 58.20 ± 10.50 | 36.46 ± 7.09 | 36.41 ± 7.30 |
> | + 4s SW inference | 60.43 ± 7.86 | 37.57 ± 7.18 | 37.71 ± 7.47 |
> | + 16s SW inference | 57.24 ± 11.00 | 36.43 ± 7.30 | 36.73 ± 7.40 |
> | + mask width = 8 | 58.12 ± 11.72 | 36.57 ±7.02 | 36.05 ± 6.74 |
> | + mask width = 16 | 57.65 ± 10.47 | 36.66 ± 7.21 | 36.92 ± 7.4 |
>
> - Results in the finetuned setting:
> |  | Test domain val | Test domain test |
> | --- | --- | --- |
> | SplashNet-Mini | 6.13 ±2.96 | 5.87 ± 3.04 |
> | + 4s SW inference | 6.10 ± 2.91 | 6.28 ± 3.09 |
> | + 16s SW inference | 5.76 ± 2.64 | 5.74 ± 2.86 |
> | + mask width = 8 | 5.85 ±2.72 | 6.02 ± 3.11 |
> | + mask width = 16 | 6.09 ± 2.88 | 5.72 ± 2.96 |

---

> > ### Comment · Reviewer_5Ux2 · 2025-08-05
> > **Thanks for the comments**
> >
> > Thanks for the response, it is great that you will include the hyperparameter search. I'm keeping my score the same because I think that there should have been more work done to show the generality of the approach; its not that hard to run this on an additional dataset for the supplement. I would still vote to accept the manuscript.

---

> > > ### Author Response · Authors · 2025-08-06
> > > **Commitment to emg2pose analyses**
> > >
> > > Thank you for the follow-up. We agree that demonstrating cross-dataset generality will strengthen the paper. Because emg2pose is a unimanual task, Split-and-Share is not applicable, but ACM and RTN can transfer directly. We will train RTN + ACM models on emg2pose and will include the full analysis, including ablations and discussion of results, in the appendix of the revised draft.

---

> > > > ### Comment · Reviewer_5Ux2 · 2025-08-07
> > > > **Thanks**
> > > >
> > > > Thanks, would be great to see if they improve emg2pose. with that happy to bump the score to a 5

---

### Official Review · Reviewer_gmaj · 2025-07-02

**Clarity:** 4
**Significance:** 3
**Originality:** 3
**Rating:** 5
**Confidence:** 5

**Summary:**

This work improves upon the emg2qwerty benchmark (https://arxiv.org/abs/2410.20081) which is the current public state-of-the-art for keyboard typing using surface electromyography. The authors propose simple and efficient architectural modifications - (1) per-session causal time normalization, (2) aggressive channel masking, (3) per-wrist encoder prior to fusing - that lead to significant improvements in this 108 user dataset on both the zero-shot user benchmark (~52% to ~36%) and finetuned benchmark (7% -> 5.5%). The authors also provide a smaller version of the model with much less params and FLOPs with spectral frequency reduction that's nearly just as performant.

**Questions:**

1. Have the authors considered entirely changing the architecture from the TDS convolutional model to transformer based?
2. For the split and share architecture, the authors can clarify the order of electrodes between the left and right bands. Is it possible to see one hand as a the mirror of another and use a "fully" shared encoder rather than learning hand specific weights?
3. Does the gain obtained from Aggressive Channel Masking indicate that a subset of channels may be more useful than others, and is there room to simply remove low SNR channels entirely for the task of typing?

**Ethical Concerns:**

["NO or VERY MINOR ethics concerns only"]

**Final Justification:**

The authors responded to the suggested improvements satisfactorily in their rebuttal. The suggestions were largely posed with the intention of further improving this work, and they shouldn't block the acceptance of this work. My rating remains unchanged.

**Limitations:**

Yes

**Quality:**

4

**Strengths And Weaknesses:**

The significance of the paper lies in the simplicity of the methodologies and the extent of the improvement obtained, especially on the zero-shot user generalization benchmark (~52% to ~36%) without any additional data.

Strong justifications are provided for each of the three contributions in the methodology via inductive priors, visualizations (peri-stimulus histograms pre and post within-session normalization), and with experimental results including detailed ablations.

Open-source code built on the original emg2qwerty repo is provided for reproducibility and continuing to improve upon this baseline.

The "mini" version of the model obtained with spectral resolution reduction achieves almost similar performance with significant reductions to model size (1/4th) and compute (0.6x).

The paper excels in overall clarity - the motivation, inductive priors for each of the improvement proposed, methodical description of the improvements, experimental results showing clear performance gains, and ablations showing how the methods stack up.

Overall, a strong paper that demonstrates an outsized improvement on a publicly available dataset and benchmark with very simple strategies for the niche but emergent modality of surface electromyography.

---

> ### Author Rebuttal · Authors · 2025-07-30
>
> We thank Reviewer gmaj for their positive assessment of our work's simplicity, significant improvements, and reproducibility, as well as their insightful questions on architectural alternatives and channel masking. Our rebuttal addresses these points, including clarifications on electrode symmetry and potential for channel ablation, which align with our focus on efficient, generalizable EMG decoding.
>
> **Q1:** *Have the authors considered entirely changing the architecture from the TDS convolutional model to transformer based?*
>
> - Though we considered changing to a transformer-based model early on, we ultimately felt that the initial hyperparameter tuning and subsequent model training would be prohibitive given our fairly limited compute budget (1x RTX 4090), and decided to instead focus on simple, principled methods starting from the already-tuned and compact TDSConv baseline, with an emphasis on efficiency and feasibility for on-device applications. We leave investigations of other block micro-architectures to future work, and note that the Split-and-Share macro-architecture is generalizable to any encoder block micro-architecture, as are the other methods we propose.
>
> **Q2:** *For the split and share architecture, the authors can clarify the order of electrodes between the left and right bands. Is it possible to see one hand as a the mirror of another and use a "fully" shared encoder rather than learning hand specific weights?*
>
> - Yes, the order of electrodes is mirror symmetric between the left and right bands, so it is possible to see one hand as a “mirror” of the other. The encoder weights are in fact fully shared between the two hands for our SplashNet models, aside from the final linear decoding layer; the reason why the last layer must differ is that the same movements correspond to different keystrokes between the left and right hands. For example, the same movement that generates an “f” keystroke on the left hand generates a “j” keystroke for the right hand. Hence, in the same sense that the right and left hands use a similar range of motions to type different sets of keys, we use the same encoder for both hands, but allow the final keystroke decoder layer to differ.
>
> **Q3:** *Does the gain obtained from Aggressive Channel Masking indicate that a subset of channels may be more useful than others, and is there room to simply remove low SNR channels entirely for the task of typing?*
>
> - We do not take the gain obtained from ACM to necessarily indicate that a subset of channels may be more useful than others, as the principle behind using ACM (breaking the model’s reliance on user-specific high-order feature correlations) still applies just as well if all channel subsets are equally informative. Nonetheless, it seems very likely true that some channels are more useful than others: different electrodes record from different muscles, which are almost certainly not uniform in their informativeness for keystroke decoding or consistency across the user population. We do not explore full-on ablations of channel subsets in this work, but we believe that this a fruitful direction, especially considering that many consumer-grade EMG devices in the near future will likely feature fewer than 16 electrodes due to cost constraints.

---

> > ### Comment · Reviewer_gmaj · 2025-08-08
> > **Thank you for the comments**
> >
> > I thank the authors for the satisfactory responses to my questions. They were largely posed with the intention of further improving this work, and they shouldn't block the acceptance of this result.

---

### Decision · Program_Chairs · 2025-09-17

**Decision:**

Accept (poster)

**Comment:**

This work offers creative and well executed ML contributions to the problem of surface typing from multivariate EMG signals in the context of neuromotor interfaces. The gains on the SoTA are significant and all reviewers endorse this work for publication at NeurIPS. This work is therefore recommended for acceptance.